# ECGluFormer: glucose prediction from ECG via multi-loss, transformer-based aggregation

Mu-Ruei Tseng
*Dept. of Computer Science and Engineering*
*Texas A&M University*
College Station, Texas
mtseng@tamu.edu

Ricardo Gutierrez-Osuna
*Dept. of Computer Science and Engineering*
*Texas A&M University*
College Station, Texas
rgutier@tamu.edu

*Abstract*—**Continuous glucose monitors (CGMs) have become ubiquitous in diabetes care but are unreliable in the hypoglycemic range, where they are most critical. We present ECGluFormer, a deep-learning (DL) model that estimates glucose levels non-invasively from single-lead ECG. Our model addresses two critical problems in ECG-based glucose prediction: (1) conventional DL models tend to under-report hypoglycemia due the rarity of those events, and (2) ECG signals can be intermittent in free-living conditions (e.g., motion artifacts, packets drops). To address these issues, ECGluFormer uses a multi-objective loss function that ensures the distribution of glucose predictions is consistent with ground-truth, and a Transformer-based model to aggregate beat-level glucose predictions when significant data losses (over 30% of all beats) are missing. We validate ECGluFormer on ambulatory data containing up to 17 days of synchronized ECG and CGM data from patients with type-1 diabetes. Our multi-objective loss function outperforms alternative loss functions across regression and classification metrics. ECGluFormer also consistently outperforms five baseline models that also aggregate beat-level predictions.**

*Index Terms*—**Diabetes, deep learning, ECG analytics, non-invasive glucose prediction, hypoglycemia.**

**Clinical relevance**. Diabetes is a chronic condition that affects nearly 600 million people worldwide. We propose a deep-learning model that can estimate glucose levels from intermittent ECG signals, avoiding the need for continuous glucose monitors, which are invasive, expensive, and notoriously inaccurate in the critical range of hypoglycemia.

## I. Introduction

A critical aspect in managing diabetes is to keep glucose levels in range (within 70-180 mg/dl). Sustained glucose above 180 mg/dl (hyperglycemia) can lead to serious long-term complications, including heart disease, kidney failure, blindness and amputation [1], [2]. Glucose levels below 70 mg/dl (hypoglycemia) are dangerous in the short-term, as they can lead to confusion, seizure, loss of consciousness, and death [3]. Though CGMs are ubiquitous, they are invasive, must be replaced periodically (every 10-15 days), and most importantly have low accuracy in the hypoglycemic range, with mean absolute relative differences (MARD) 2-3 times higher than those in the normoglycemic range (MARD<10%) according to FDA approval letters for the four leading selling CGMs (Abbott Freestyle Libre 3: 13.4-18.8%; Dexcom G7 (16.0-27.8%; Medtronic 780G: 14.9-19.4%; Senseonics Eversense

E3: 13.0-19.5%). Further, CGMs are susceptible to temporal shifts and compression artifacts, especially at night when counter-regulatory responses are attenuated [4]. Finally, CGM accessibility is limited due to socioeconomic disparities or insurance barriers [5], [6]. Therefore, there is **a critical need** for new sensing techniques that can monitor glucose levels continuously and non-invasively.

Several physiological variables may be used as indirect indicators of hypoglycemia [7], in particular changes in cardiac signals such as time-domain information, e.g., heart rate (HR) and heart rate variability (HRV) [8], and morphological information at the beat level, e.g., a lengthened QT interval [9]. HRV measurements are appealing since they can be estimated from photoplethysmography (PPG). However, PPG is notoriously sensitive to motion artifacts, particularly when measured at the wrist (e.g., smartwatches). In contrast, ECG is robust to motion artifacts and provides fine-grained information about beat morphology.

This paper addresses three critical issues in ECG-based glucose prediction. *First*, there is a large imbalance in the distribution of glucose levels (less than 4% are in the hypoglycemic range). As a result, models trained on conventional loss functions (i.e., mean squared error) tend to over-predict in the hypoglycemic range. *Second*, when recorded in free-living conditions, ECG experience data losses due to wireless connectivity issues and changes in the electrode-skin interface. As a result, glucose estimation models must be able to operate when sensor data is intermittent. *Finally*, framing the problem as one of binary classification (i.e., hypo- vs. eu-glycemia) is problematic given that GCMs are inaccurate in the hypoglycemic range.

To address these issues, we propose **ECGluFormer**, a two-stage deep learning (DL) framework for estimating *glucose levels* (i.e., a regression rather than a binary classification task) from ambulatory ECG. ECGluFormer combines a beat-level encoder that captures morphological information with a transformer that incorporates time-based positional encodings to model the relative timing of each heartbeat. This design allows the model to learn physiologically meaningful temporal patterns (e.g., HRV) even in the presence of significant data losses (over 30% missed beats). To address the distribution imbalance, we introduce a composite loss function that jointly

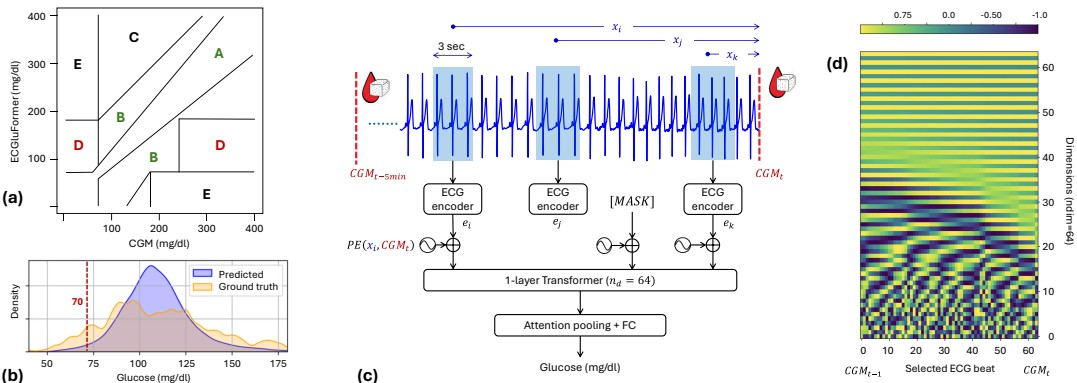

Fig. 1. **(a)** Clarke Error Grid. Zone A: clinically accurate; B: clinically acceptable; C: over-correcting; D: *failure to detect*; E: erroneous treatment. **(b)**. Mismatch in the distribution of blood glucose (yellow), and predictions using the MSE loss. **(c)** ECGluFormer architecture. An ECG encoder based on InceptionTime consumes 3-sec ECG windows to predict the next glucose level $CGM_t$. Each 3-sec ECG embedding is treated as a token with time-based positional encoding. A 1-layer Transformer encoder processes tokens and aggregates them via attention pooling. **(d)** Visualization of time-based positional encodings for ECG beats within a CGM segment. The y-axis represents individual ECG beats, sorted by time *relative* to the CGM timestamp, while the x-axis corresponds to the embedding dimensions ($n_{\dim} = 64$). This allows the model to incorporate precise timing information rather than an index in the sequence.

penalizes deviations in the distribution of predictions and per-sample errors, with greater emphasis on lower glucose values. The main contributions of this paper are:

- An encoder model for short ECG segments (3 seconds) that uses a composite loss function to jointly optimize the distributional alignment of glucose predictions against ground truth and per-sample accuracy, to address data imbalances in the hypoglycemic region.
- A transformer model that aggregates information from intermittent sampled ECG embeddings through a time-based positional encoding that accounts for the relative timing of non-uniformly sampled heart beats.
- Model evaluation on regression (predicting glucose levels) and classification tasks (hypo- vs. normo-glycemia).

## II. RELATED WORK

### A. Physiological indicators of hypoglycemia

Several physiological variables have been investigated as potential indirect indicators of hypoglycemia [7]. Early work focused on skin temperature and skin conductivity, which decrease at the onset of hypoglycemia [10]. Several commercial devices were developed in the 1980s [11] but showed high false-alarms rates (3:1) due to perspiration unrelated to hypoglycemia [12], so they never received FDA approval. Electroencephalography (EEG) has also been used as a potential indicator of hypoglycemia. Early work [13], [14] showed an association between hypoglycemia and increases in $\delta$ and $\gamma$ bands and decreases in $\alpha$ frequency. More recent work shows an association with a decrease in signal complexity [15]. However, EEG measurements are far more involved than skin temperature/conductivity, so at present ECG is impractical for hypoglycemia detection in free-living settings.

### B. ECG-based glucose prediction

Most of the work on non-invasive hypoglycemia detection has focused on ECG. Early work focused on extracting physiologically relevant features (e.g., QRS duration, QTc interval,

T-wave amplitude) [16], [17] and feeding them to statistical machine learning (ML) models. While these methods offer some interpretability, their performance heavily depends on accurate detection of fiduciary points, which is often not guaranteed in noisy ECG ambulatory recordings. To mitigate these issues, recent studies have used DL to learn latent representations directly from raw or minimally preprocessed ECG signals. For example, Porumb et al. [18] proposed a CNN-based beat encoder followed by recurrent layers to aggregate information across time. Other models [19] use CNNs and bidirectional LSTMs to process longer ECG sequences for broader classification tasks.

A key limitation of both traditional and DL methods is the lack of attention to temporal alignment between ECG beats and glucose measurements. Fixed-size beat sampling can obscure the true timing of physiological events relative to glucose dynamics, especially under irregular heart rates or in noisy environments. Few models explicitly account for the variable intervals between beats or leverage their temporal context as a predictive signal. *Our model address these gaps by modeling beat timing explicitly using positional encodings within a transformer architecture. This approach enables finer-grained temporal reasoning and also improves robustness to sampling irregularities.*

### C. Transformer models

First introduced for NLP tasks [20], Transformers have gained traction in physiological signal modeling due to their ability to capture long-range dependencies and contextual relationships without relying on recurrence. Several Transformer models have been applied to ECG analysis—primarily in arrhythmia classification. For example, ECG-DETR [21] combines a CNN backbone with a Transformer encoder-decoder to detect and classify heartbeats from short ECG segments. ECGTransForm [22] enhances temporal and spatial features via multi-scale CNNs and channel recalibration before a BiTransformer encoder. CAT-Net [23] integrates attention

layers between convolutional blocks and passes their output into a transformer encoder to classify arrhythmias.

While effective for short-term diagnostic tasks (e.g., beat classification or arrhythmia detection), these models operate on short, fixed-length ECG segments, typically 5–10 sec long. *To our knowledge ours is the first effort to apply Transformers to minute-scale glucose prediction from ECG by treating ECG beats as time-aware tokens. This enables interpretable and scalable modeling of glucose dynamics using transformer-based architectures without the computational burden of processing raw waveforms over long windows.*

## III. METHODS

Illustrated in Figure 1c, ECGluFormer consists of two key building blocks: an encoder that consumes 3-sec windows of ECG to produce a glucose prediction, and a Transformer that aggregates ECG embeddings over a 5-min period to produce a final estimate of glucose levels. Before feeding data to the encoder, we pre-process the ECG signals using NeuroKit2 [24], which performs signal denoising and R-peak detection. For each detected R-peak, we extract a fixed 3-sec window centered on R peaks to capture the morphology of individual heart beats. Each beat is then aligned with the nearest future CGM measurement to assign a corresponding glucose label. Finally, we apply z-score normalization to each extracted ECG beat to remove inter-beat amplitude variability and allow the encoder to focus on shape-based features.

### A. Beat-level ECG encoder

Our encoder uses a backbone based on InceptionTime [25], a 1D CNN that captures multi-scale temporal patterns through parallel convolutional branches with varying kernel sizes. As illustrated in Figure 1b, however, the probability density of glucose levels for an individual is significantly lower in the critical hypoglycemic range (<70 mg/dl). As a result, models trained to minimize the mean-squared-error (MSE) tend to under-report in that range. To address this issue, ECGluFormer uses a multi-objective loss function that combines a distribution loss (DistLoss) [26], and a Normalized MSE (NMSE) [27] term to minimize the per-sample prediction error:

$$
\mathcal{L}(\hat{G}_{\text{batch}}, G_{\text{batch}}) = \underbrace{\text{MSE}(\mathcal{S}_{\hat{G}_{\text{batch}}}, \tilde{G}_{\text{pseudo}})}_{\text{distribution loss}} \\
+ \lambda \cdot \underbrace{\text{NMSE}(\hat{G}_{\text{batch}}, G_{\text{batch}})}_{\text{Sample-wise loss}}, \tag{1}
$$

where $\lambda$ is a hyperparameter that balances both terms, $G_{\text{batch}} = \{g_1, \ldots, g_n\}$ are the ground truth glucose values in the batch, $\hat{G}_{\text{batch}} = \{\hat{g}_1, \ldots, \hat{g}_n\}$ are the corresponding model predictions, $\tilde{G}_{\text{pseudo}}$ is the set of pseudo-labels constructed by replicating each label $g_i$ according to its expected frequency in the batch (estimated via KDE over the label distribution), $\mathcal{S}_{\hat{G}_{\text{batch}}}$ is the sorted sequence of predictions, representing the prediction distribution, $\mathcal{L}_{\text{dist}}$ measures the discrepancy (e.g., via MSE) between the sorted predictions and pseudo-labels, both normalized to focus on distributional shape rather

than scale, and NMSE is computed on the original (non-normalized) glucose values to maintain numerical stability and avoid division by near-zero values:

$$
\text{NMSE}(\hat{G}, G) = \frac{1}{n} \sum_{(\hat{g}, g) \in (\hat{G}, G)} \frac{\|\hat{g} - g\|^2}{\|g\|^2} \times \frac{1}{f_g}, \tag{2}
$$

Additionally, we apply an inverse-frequency $\frac{1}{f_g}$ weighting based on the distribution of ground truth glucose values to over-penalize prediction errors in low-probability regions, encouraging the model to perform robustly across the full glucose range. This composite loss enables the model to align its output distribution with the ground truth while preserving accuracy at the sample level, which is particularly beneficial under imbalanced data conditions.

### B. CGM-level glucose prediction

We use a Transformer model to aggregate time-stamped beat-level ECG information at the CGM level (5-min intervals). Unlike conventional approaches that treat all beats uniformly, our model explicitly accounts for the *temporal misalignment* between ECG beats and CGM measurements. Since the number of ECG beats within a CGM segment varies with heart rate, we randomly sample $k$ beats without replacement from each segment ($k = 64$) and sort them according to their recorded time.

To capture the temporal context of each beat relative to the CGM reading, we incorporate a *time-based positional encoding* based on each beat's timestamp offset from the CGM time, rather than using simple sequential indices (e.g., 0, 1, 2...). This allows the model to learn from the actual timing of events rather than their order in the input sequence. Let $x_i$ denote the $i$-th ECG token, where each token represents a 3-sec ECG window, and let $t(x_i)$ be the timestamp at the center of that window. Let $t(\text{CGM}_t)$ denote the timestamp of the CGM reading at prediction time $t$. We define the relative position of each token as:

$$
\text{pos}_i = t(\text{CGM}_t) - t(x_i) \tag{3}
$$

To encode this timing information, we use the standard sinusoidal positional encoding scheme [20]. An example of this positional encoding is shown in Figure 1d.

Since we use a 5-minute prediction window and each ECG token spans 3 seconds, the input sequence consists of approximately 100 ECG tokens. Each $\text{pos}_i \in [0, 300]$ is measured in seconds and encoded using sinusoidal positional encoding to retain timing information relative to the prediction target. This design allows the Transformer to contextualize each beat embedding based on its relative timing, enabling the model to attend more effectively to temporally relevant beats. Its attention mechanism captures inter-beat dependencies, while the positional encoding grounds the model in the temporal structure of the ECG-CGM alignment. Furthermore, unlike NLP models where tokenizers provide fixed embeddings for each word, our ECG encoder is not frozen. This allows the transformer to refine the beat-level embeddings during training to improve glucose prediction performance.

To improve generalization and reduce overfitting, we adopt a BERT-style masking strategy [28] during training. Specifically, 10% of the input ECG tokens, each representing a 3-sec segment, are randomly selected and replaced with a learnable `[MASK]` embedding. This encourages the model to infer information from surrounding context, increasing robustness to missing or noisy beats. Each sequence of embeddings is augmented with time-based sinusoidal positional encoding and normalized using LayerNorm. The resulting sequence is then passed through a single-layer Transformer encoder, producing contextualized token representations $h_t \in R^d$, where $t = 1, \ldots, T$ indexes the tokens. A second LayerNorm is applied after the transformer to stabilize training.

To aggregate information across the sequence, we apply attention pooling, computing attention weights:

$$\alpha_t = \frac{\exp(\mathbf{w}^\top \mathbf{h}_t)}{\sum_{t'=1}^{T} \exp(\mathbf{w}^\top \mathbf{h}_{t'})} \tag{4}$$

and pooled representation:

$$\mathbf{z} = \sum_{t=1}^{T} \alpha_t \mathbf{h_t} \tag{5}$$

Unlike average pooling, attention pooling enables the model to focus on more informative or salient beats, producing a context-aware summary that better reflects the predictive features relevant to glucose dynamics. This summary vector $\mathbf{z}$ is passed through a feedforward regressor to predict the target glucose value. The same objective function introduced in the previous section is used to train the ECGluFormer.

## IV. IMPLEMENTATION

### A. Model training

The experiments and evaluations in this paper are conducted on the PhysioCGM dataset [29], which contains synchronized ECG (Zephyr Bioharness) and CGM (Dexcom G6) recordings from 10 participants who wore both devices for up to 17 days.

We implemented ECGluFormer in PyTorch. Due to large inter-individual differences, we developed a separate model for each subject. We used a batch size of 512 for to train the ECG encoder and 50 for the CGM-level aggregator. The ECG encoder uses an InceptionTime backbone with 16 filters. The CGM-level aggregator uses a single-layer encoder-only Transformer with 4 attention heads, followed by two fully connected (FC) layers of size 64 and 1, respectively. A ReLU activation and dropout (rate = 0.2) are applied between the layers. Both models are optimized using AdamW with a weight decay of 0.01. The learning rates are set to 0.0005 for the ECG encoder and 0.0001 for the CGM-level model. We apply learning rate decay with a factor of 0.5 if the validation loss does not improve for 20 consecutive epochs. Models were trained on NVIDIA RTX 3090 GPUs.

To simulate the effect of missing ECG data, we randomly sample 64 non-overlapping 3-sec ECG beats per epoch during training, **which amounts to 34% data loss over a 5-min CGM period**. This randomized sampling forces the Transformer to *extract dynamic information from intermittent beats*, and introduces natural variability across epochs, effectively serving as a form of data augmentation that enhances model robustness. To ensure a balanced evaluation, we split the dataset into training, validation, and test sets in a 70:15:15 ratio using stratified sampling to preserve the proportion of hypoglycemic events. We perform hyperparameter tuning on the validation set, and report results on the test set. During testing, we apply the same sampling strategy, and the reported performance metrics are averaged over 20 independent test runs to ensure robustness.

### B. Evaluation metrics

We evaluate model performance using quantitative metrics of glucose prediction and hypoglycemia prediction:

- **Clarke Error Grid (CEG)**: Proportion of predictions in Zone A+B (clinically accurate/acceptable) and Zone D (failure to detect).
- **Root Mean Square Error (RMSE)**: Average prediction error between predicted and ground truth glucose values.
- **Hypoglycemia classification**: Glucose predictions are converted into pseudo-probabilities using min-max normalization (40 to 180 mg/dL). Ground truth glucose values are binarized (1: below 70 mg/dL; 0: otherwise). We report sensitivity and specificity at the EER threshold (false positive rate = false negative rate).

This combination of metrics offers a comprehensive assessment of both clinical safety and numerical accuracy.

## V. RESULTS

### A. Encoder capacity

We performed hyperparameter tuning for the ECG encoder with $n_f \in 4, 8, 16$, which yield output embedding dimensions of 16, 32, and 64, respectively. Increasing the number of filters enables the encoder to capture more complex temporal patterns, potentially improving predictive performance at the cost of higher computational complexity. An encoder with $n_f = 16$ (embedding size $n_d = 64$) achieves the best overall performance (results not shown due to page limits), and was used for all subsequent experiments. Figure 2 shows a t-SNE projection of the ECG embeddings. Each point corresponds to an encoded ECG beat, color-coded by the ground truth glucose values (Figure 2a) and heart rate (Figure 2b). We observe a distinct left-to-right gradient from low to high glucose levels in the t-SNE embedding but not for heart rate, indicating that the ECG encoder is able to disentangle glucose information from heart rate, which otherwise is more dominant.

### B. Loss functions

We evaluated our proposed loss function mean-squared-error (MSE), normalized MSE (i.e., percent error w.r.t. ground truth) [27], and Distribution Loss (DistLoss) on NMSE [26]. Results are illustrated in Figure 2c and summarized in Table I. Our combined loss shows the lowest percentage in Zone D (*failure to detect*) –the most critical region in the

TABLE I
EVALUATION OF LOSS FUNCTIONS ON CLARKE ERROR GRID MEASURES (A+B, D), HYPOGLYCEMIA CLASSIFICATION (SENSITIVITY AND SPECIFICITY AT EER), AND GLUCOSE PREDICTION (RMSE)

| Metrics | Model Type | Subject | | | | | | | | |
|---|---|---|---|---|---|---|---|---|---|---|
| | | c1s01 | c1s03 | c1s05 | c2s01 | c2s02 | c2s03 | c2s04 | c2s05 | Average |
| Zone A+B (%) ↑ | MSE | 94.37 | 96.60 | 98.10 | 96.89 | 97.59 | 98.95 | 97.92 | 95.28 | 96.96 |
| | NMSE | 94.94 | 96.88 | 98.24 | 97.67 | 98.79 | 99.04 | 98.64 | 96.15 | 97.54 |
| | Dist Loss (NMSE) | 95.05 | 97.18 | 98.40 | 97.69 | 98.39 | 99.19 | 98.47 | 96.32 | 97.58 |
| | **Ours** | 96.58 | 98.11 | 98.91 | 98.19 | 99.13 | 98.89 | 99.08 | 97.39 | 98.28 |
| Zone D (%) ↓ | MSE | 5.63 | 3.39 | 1.90 | 3.11 | 2.41 | 1.05 | 2.08 | 4.72 | 3.04 |
| | NMSE | 5.06 | 3.11 | 1.76 | 2.31 | 1.21 | 0.95 | 1.35 | 3.85 | 2.45 |
| | Dist Loss (NMSE) | 4.94 | 2.82 | 1.60 | 2.29 | 1.61 | 0.80 | 1.50 | 3.68 | 2.41 |
| | **Ours** | 3.37 | 1.87 | 1.09 | 1.59 | 0.87 | 0.56 | 0.80 | 2.60 | 1.59 |
| Sensitivity ↑ | MSE | 0.65 | 0.71 | 0.79 | 0.74 | 0.87 | 0.79 | 0.76 | 0.75 | 0.76 |
| | NMSE | 0.70 | 0.72 | 0.82 | 0.79 | 0.91 | 0.80 | 0.84 | 0.76 | 0.79 |
| | Dist Loss (NMSE) | 0.70 | 0.73 | 0.83 | 0.80 | 0.90 | 0.85 | 0.82 | 0.77 | 0.80 |
| | **Ours** | 0.70 | 0.76 | 0.84 | 0.80 | 0.92 | 0.80 | 0.86 | 0.74 | 0.80 |
| Specificity ↑ | MSE | 0.58 | 0.67 | 0.75 | 0.69 | 0.86 | 0.77 | 0.73 | 0.71 | 0.72 |
| | NMSE | 0.64 | 0.69 | 0.79 | 0.76 | 0.89 | 0.78 | 0.82 | 0.73 | 0.76 |
| | Dist Loss (NMSE) | 0.66 | 0.69 | 0.80 | 0.76 | 0.88 | 0.82 | 0.81 | 0.74 | 0.77 |
| | **Ours** | 0.66 | 0.73 | 0.80 | 0.77 | 0.89 | 0.79 | 0.84 | 0.72 | 0.77 |
| RMSE ↓ | MSE | 23.53 | 22.52 | 17.08 | 19.73 | 15.65 | 18.66 | 22.08 | 19.34 | 19.82 |
| | NMSE | 25.59 | 24.93 | 17.59 | 21.71 | 17.51 | 24.13 | 23.26 | 23.00 | 22.21 |
| | Dist Loss (NMSE) | 24.81 | 25.07 | 18.21 | 21.93 | 18.19 | 20.32 | 23.61 | 20.76 | 21.61 |
| | **Ours** | 30.93 | 29.64 | 19.94 | 29.81 | 19.36 | 31.62 | 30.54 | 25.49 | 27.17 |

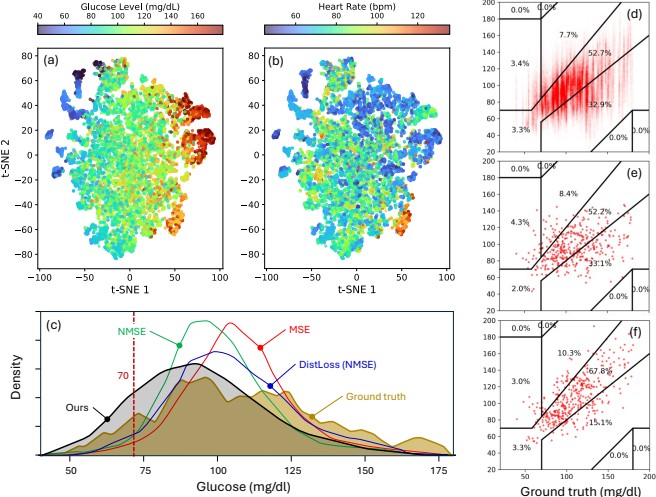

Fig. 2. t-SNE plots of embeddings from the ECG encoder on subject c2s02. Each point represents a 3-sec ECG segment, color-colored according to (a) ground-truth glucose levels and (b) heart rate. (c) Distribution of ground truth glucose (orange) vs. glucose distributions for four los functions: MSE, NMSE, DistLoss on MSE, and ours (inverse frequency-weighted DistLoss on MSE). (d-f) Clarke-Error Grid for beat-level predictions, baseline B2 and ECGGluFormer, respectively.

Clarke Error Grid, and the highest percentage in Zones A+B (*clinically accurate+acceptable*). We do not report results in Zones C and E (in most cases, below 1%). Further, a two-way ANOVA on Zone D percentages using subject identity and loss function as independent factors shows statistical significance ($p_{\text{subject}}, p_{\text{loss}} < 0.001$). However, because each subject contributes multiple entries (one per loss), the assumption of independent observations may be violated. Hence, we also performed a non-parametric Friedman test across loss functions, which confirmed a significant effect ($p = 0.0001$), supporting the robustness of our findings.

While the MSE loss achieves the lowest RMSE, it also leads to the highest %-age in Zone D (*failure to detect*), as well as the lowest sensitivity for hypoglycemia detection. This occurs because MSE-trained models tend to regress toward the population mean. Introducing the DistLoss with MSE improves performance in this critical region, though at the cost of a modest increase in RMSE. Our proposed loss function further enhances performance in the hypoglycemic range by replacing MSE with Normalized MSE (NMSE), which treats errors in the lower glucose range as more significant. Additionally, incorporating inverse frequency weighting further emphasizes rare but clinically important cases by penalizing prediction errors in low-probability regions more heavily.

### C. Comparison against baselines

We are not aware of any open-source implementations for ECG-based glucose prediction. Hence, we re-implemented two representative baseline models for comparison:

- **B1: Feature engineering**. B1 uses Neurokit2 to extract six ECG features: P and T wave durations, QRS duration, QTc interval, and HR. Features were averaged over each 5-minute interval and fed to a regression pipeline [17].
- **B2: Feature learning**: B2 is the CNN-LSTM of [18] for nocturnal hypoglycemia detection from 1-sec ECG segments. To align it with our task, we modified the model to allow glucose regression but kept the original 1-sec analysis window. To ensure a fair comparison, we trained B2 using our proposed loss function.

We also evaluated the Transformer model in ECGGluFormer against three alternative aggregation strategies:

TABLE II
COMPARISON OF ECGLUFORMER VS. BASELINES ON THE SAME METRICS OF TABLE I.

| Metrics | Model Type | Subject | | | | | | | | Average |
|---|---|---|---|---|---|---|---|---|---|---|
| | | c1s01 | c1s03 | c1s05 | c2s01 | c2s02 | c2s03 | c2s04 | c2s05 | |
| Zone A+B (%) ↑ | Feature-based [17] | 94.41 | 95.09 | 95.78 | 95.45 | 95.81 | 98.13 | 96.94 | 93.29 | 95.61 |
| | Porumb et al. [18] | 95.67 | 96.40 | 97.64 | 96.59 | 96.04 | 98.86 | 97.72 | 94.15 | 96.63 |
| | Average | 96.97 | 97.85 | 98.77 | 98.36 | 99.03 | 99.13 | 99.38 | 97.52 | 98.37 |
| | Histogram + FC | 96.95 | 97.16 | 98.77 | 97.71 | 99.03 | 99.19 | 99.31 | 97.48 | 98.20 |
| | LSTM | 96.75 | 98.22 | 99.00 | 98.28 | 98.90 | 99.34 | 98.97 | 98.30 | 98.47 |
| | **ECGluFormer** | 97.07 | 98.96 | 99.75 | 98.71 | 99.60 | 99.29 | 99.22 | 99.29 | 98.99 |
| Zone D (%) ↓ | Feature-based [17] | 5.59 | 4.91 | 4.22 | 4.55 | 4.19 | 1.87 | 3.06 | 6.71 | 4.39 |
| | Porumb et al. [18] | 4.33 | 3.60 | 2.36 | 3.41 | 3.96 | 0.95 | 2.28 | 5.85 | 3.34 |
| | Average | 3.03 | 2.15 | 1.23 | 1.64 | 0.97 | 0.63 | 0.62 | 2.48 | 1.60 |
| | Histogram + FC | 3.05 | 2.84 | 1.23 | 2.29 | 0.97 | 0.71 | 0.69 | 2.52 | 1.79 |
| | LSTM | 3.25 | 1.78 | 1.00 | 1.70 | 1.10 | 0.62 | 1.03 | 1.70 | 1.52 |
| | **ECGluFormer** | 2.93 | 1.04 | 0.25 | 1.29 | 0.40 | 0.59 | 0.78 | 0.71 | 1.00 |
| Sensitivity ↑ | Feature-based [17] | 0.88 | 0.71 | 0.72 | 0.89 | 0.77 | 0.75 | 0.65 | 0.64 | 0.75 |
| | Porumb et al. [18] | 0.72 | 0.53 | 0.86 | 0.72 | 0.81 | 0.78 | 0.75 | 0.63 | 0.72 |
| | Average | 0.72 | 0.77 | 0.85 | 0.88 | 0.96 | 0.82 | 0.91 | 0.81 | 0.84 |
| | Histogram + FC | 0.73 | 0.78 | 0.85 | 0.90 | 0.96 | 0.82 | 0.90 | 0.81 | 0.84 |
| | LSTM | 0.65 | 0.80 | 0.88 | 0.82 | 0.91 | 0.78 | 0.89 | 0.81 | 0.82 |
| | **ECGluFormer** | 0.71 | 0.86 | 0.94 | 0.85 | 0.94 | 0.87 | 0.92 | 0.84 | 0.86 |
| Specificity ↑ | Feature-based [17] | 0.34 | 0.69 | 0.65 | 0.36 | 0.74 | 0.50 | 0.64 | 0.53 | 0.56 |
| | Porumb et al. [18] | 0.61 | 0.52 | 0.83 | 0.71 | 0.79 | 0.75 | 0.72 | 0.55 | 0.68 |
| | Average | 0.69 | 0.75 | 0.84 | 0.83 | 0.90 | 0.80 | 0.89 | 0.73 | 0.80 |
| | Histogram + FC | 0.69 | 0.74 | 0.83 | 0.82 | 0.89 | 0.79 | 0.89 | 0.74 | 0.80 |
| | LSTM | 0.57 | 0.76 | 0.87 | 0.78 | 0.90 | 0.77 | 0.85 | 0.74 | 0.78 |
| | **ECGluFormer** | 0.68 | 0.84 | 0.92 | 0.82 | 0.93 | 0.86 | 0.90 | 0.82 | 0.85 |
| RMSE ↓ | Feature-based [17] | 26.62 | 26.43 | 20.68 | 24.88 | 22.43 | 25.33 | 23.02 | 25.78 | 24.40 |
| | Porumb et al. [18] | 31.13 | 31.33 | 19.33 | 27.96 | 24.63 | 32.80 | 32.72 | 30.39 | 28.79 |
| | Average | 26.89 | 26.74 | 17.29 | 25.39 | 16.94 | 26.87 | 26.10 | 22.03 | 23.53 |
| | Histogram + FC | 25.09 | 22.32 | 15.97 | 23.52 | 14.12 | 25.77 | 23.39 | 20.36 | 21.32 |
| | LSTM | 26.69 | 23.96 | 17.85 | 23.36 | 15.81 | 24.35 | 25.77 | 27.34 | 23.14 |
| | **ECGluFormer** | 22.13 | 22.05 | 15.43 | 24.02 | 13.09 | 22.76 | 23.65 | 23.49 | 20.83 |

- **B3: Ensemble averaging**: B3 computes the average glucose prediction from the ECG encoder over the 5-minute CGM sampling period.
- **B4: Stacked generalization**: B4 computes the histogram of glucose predictions at 1 mg/dl resolution over the 5-min window, and passes it to a fully-connected network to generate a single glucose prediction [30].
- **B5: LSTM**: B5 uses a 1-layer LSTM (with hidden dimension equal to the transformer's $d_{model}$) to consume the sequence of ECG embeddings, and the same fully connected prediction head as ECGluFormer to generate the final glucose prediction over the 5-min window.

Results are summarized in Table II. ECGluFormer consistently outperforms the five baselines for all metrics. It yields the highest average percentage in Zone A+B (98.99%) and the lowest Zone D error rate (1.00%), indicating better clinical accuracy and fewer critical prediction failures. ECGluFormer also achieves the best overall sensitivity (0.86), while maintaining the lowest RMSE (20.83) among all models. These results highlight the advantage of transformer-based beat-level modeling over both feature-engineered and end-to-end CNN-LSTM baselines. We conducted the same statistical analysis to evaluate the effect of model type on the performance in Zone D percentages. A two-way ANOVA shows significant effects for both subject and model ($p_{subject}, p_{model} < 0.001$), The Friedman test further confirmed a significant difference in performance ranks across models ($p < 0.0001$).

## VI. DISCUSSION

We have presented ECGluFormer, a DL model that estimates glucose levels from ECG recordings in free-living conditions. ECGluFormer combines two innovations, (1) a multi-objective loss function that minimizes distributional differences between ground truth and predicted glucose and penalizes errors in the hypoglycemic range at the beat level, and (2) a Transformer architecture that aggregates beat-level predictions while preserving temporal information in the presence of missed beats (36% in our experiments).

We compared our combined loss function against the conventional MSE loss, the normalized MSE loss, and the DistLoss [26]. Our results show that neither the NMSE or DisLoss are able to compensate for the large imbalance in the hypoglycemic range, and that an additional loss term (inverse frequency; IF) is critical. When trained with our loss function (NMSE + DistLoss + IF), the ECG encoder extracts beat-morphology information that captures glucose variability despite the more dominant information: heart rate.

We validated ECGluFormer against five baselines: (B1) a traditional feature-engineering approach that extracts fiduciary points in the ECG signal, (B2) a SOTA CNN-LSTM model for ECG hypoglycemia detection [18], (B3) an ensemble averaging method, (B4) a stacked-generalization method that preserves the distribution of glucose predictions, and (B5) an LSTM that also preserves ordering. ECGluFormer outperforms the five baselines in terms of hypoglycemia detection (zones D

in the Clarke Error Grid, sensitivity and specificity at EER) and glucose prediction (RMSE). Note that baselines B3-5 use the same ECG encoder as ECGluFormer. Thus, ECGluFormer's superiority can only be attributed to the fact that it encodes the relative timing of ECG embeddings, likely preserving dynamic ECG information that is associated with hypoglycemia (e.g., HRV.) Critically, ECGluFormer can capture this information even when a high proportion of the beats is missing –time-frequency HRV analysis is not feasible in such cases.

## VII. LIMITATIONS AND FUTURE WORK

A limitation of this study is the relatively small dataset. To our knowledge, however, PhysioCGM is the largest publicly available dataset containing synchronized ECG-CGM recordings. Due to the dataset size, we have not yet evaluated its generalization properties across subjects. Based on our experience with other CGM-related studies, building subject-independent models requires a minimum of 20+ subjects. Transfer learning may be a good option to address this challenge, using pretrained foundation models for ECG (e.g., ECG-FM, HuBERT-ECG) and then fine-tuning on our glucose-prediction task. Our two-stage design with separate beat-level encoder and Transformer aggregator makes it straightforward to swap in a pretrained backbone without modifying the aggregation head. Further, we can think of two adaptation strategies given the current amount of data: (1) *Leave-one-subject-out (zero-shot)*: train ECGluFormer on N–1 subjects (using a foundation-initialized encoder + Transformer) and directly test on the held-out participant with no further tuning to measure pure cross-patient performance; (2) *Few-shot fine-tuning*: fine-tune the entire pipeline (encoder and aggregation head) on a small amount of subject-specific ECG–CGM data to assess how quickly it personalizes to new patients.

ECGluFormer is a lightweight model that contains 410k parameters and achieves an average inference time of 8.2 milliseconds per 5-minute CGM segment on an NVIDIA RTX 3090 GPU, making the model well-suited for real-time inference. Future work will examine implementation on edge devices (e.g., smartphone chipsets.)

## VIII. CONCLUSION

Our results show that non-invasive glucose prediction with consumer-grade ECG devices in free-living conditions and in the presence of sensor failures is feasible. This requires aggregating morphological information from ECG beat morphology while preserving the relative timing of ECG signals at the time scale of glucose dynamics.

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
