# OpenReview forum: "ECGluFormer: glucose prediction from ECG via multi-loss, transformer-based aggregation"
_IEEE.org/EMBS/BHI/2025/Conference — BHI 2025_

### Official Review · Reviewer_oyMx · 2025-07-05
**Well written paper, however, there are several technical and scientific aspects to address.**

**Confidence:** 5
**Clarity Of Writing:** great
**Clinical Significance:** fair
**Methodological Novelty:** fair
**Overall Rating:** 4
**Final Rating:** 4

**Experiments And Results:**

fair

**Questions For The Authors:**

Given the subject-specific modeling approach, have you evaluated subject-independent models or transfer-learning schemes? What is the performance degradation relative to the personalized models presented?

Can you provide a subject-by-subject Clarke Error Grid (CEG) for the different loss functions compared in Table I? A visual CEG plot for each loss function, rather than just zone percentages, would help clarify the specific trade-offs between RMSE and clinical risk (e.g., are the errors in Zone D near the border of B, or are they severe failures?).

There appears to be an RMSE discrepancy. In Table I, your proposed loss yields the highest RMSE (27.17), but in Table II, the full ECGluFormer model achieves the lowest RMSE (20.83). Could you clarify the reason for this significant difference?

**Strengths:**

- The work effectively addresses the well-documented inaccuracy of commercial CGMs during hypoglycemia and the clear clinical need for reliable, non-invasive monitoring methods.

- The two core contributions are well-conceived. The composite loss function directly tackles the critical challenge of data imbalance in the hypoglycemic range. The Transformer with time-aware tokens elegantly captures dynamic information like heart-rate variability even with irregular sampling and high data loss, outperforming strong baselines that use the same encoder.

- The paper includes a thorough evaluation with clear ablation studies on loss functions, qualitative t-SNE visualizations showing feature disentanglement, and comparisons against five relevant baselines. The use of statistical tests to confirm the significance of the results adds to the paper's rigor.

- The authors provide key implementation details, including model hyperparameters, batch sizes, optimizer settings, and hardware used, which is valuable for reproducibility.

**Summary Of The Paper:**

The paper presents ECGluFormer, a two-stage deep-learning framework that estimates blood-glucose concentration non-invasively from single-lead ECG recorded in free-living conditions. Stage 1 is an InceptionTime-based beat encoder that predicts glucose for every 3-second ECG snippet. A composite loss of combining a Normalized MSE with a Distribution Loss and inverse-frequency weighting is used to penalize errors more heavily in the hypoglycemic range and enforce distributional alignment between predictions and the ground truth.
Stage 2 aggregates these beat-level embeddings over each 5-minute interval using a 1-layer Transformer. This stage uses time-offset positional encodings to preserve the irregular timing between heartbeats and is trained with BERT-style masking to tolerate significant data loss (over 30% of beats).
Experiments were conducted on the PhysioCGM dataset, which includes 10 participants with T1D, each with up to 17 days of data. Using subject-specific models, ECGluFormer was compared against five baselines and achieved the best mean performance: 98.99% of points in Clarke Error Grid (CEG) zones A+B, 1.00% in zone D, 0.86 sensitivity, and an RMSE of 20.83 mg/dL.

**Weaknesses:**

- Poor Choice of Ground Truth: The fundamental weakness is the reliance on CGM data as the ground truth. The paper itself notes that CGMs are "notoriously inaccurate in the critical range of hypoglycemia". Using an imperfect and error-prone measurement as the "gold standard" fundamentally limits the potential clinical validity of any model trained on it and may lead the model to learn the CGM's error patterns rather than the true physiological state.

- Limited Generalizability: The study is limited to a small cohort of 10 participants, and a separate model is trained for each subject. No cross-subject or leave-one-out validation is reported, leaving the model's generalizability to new, unseen users completely unknown.

- The practice of aligning each beat with the "nearest future CGM measurement" creates a risk of label leakage. Information from the prediction target (the future glucose value) might be present within the 3-second ECG window, which could artificially inflate performance metrics.

- The related work section is narrow, it could leverage other literature on noninvasive glucose measuring.

- The reported average RMSE of ~21 mg/dL exceeds the accuracy requirements of clinical standards like ISO 15197 for blood glucose meters. The paper does not benchmark its results against such established clinical thresholds.

- Synthetic Data Loss: The simulation of missing data involves randomly dropping a fixed percentage of beats. It is unclear how closely this synthetic process mimics real-world signal loss from motion artifacts or R-peak detection failures

---

### Official Review · Reviewer_Egon · 2025-07-16

**Confidence:** 3
**Clarity Of Writing:** good
**Clinical Significance:** great
**Methodological Novelty:** good
**Overall Rating:** 7
**Final Rating:** 7

**Experiments And Results:**

great

**Questions For The Authors:**

* The model appears computationally intensive; a discussion on real-world deployment feasibility, including latency and device compatibility, should be included.
* Additional evaluation on external or publicly available ECG-CGM datasets (if available) is necessary to validate generalizability.
* Incorporate interpretability studies to identify ECG features contributing to glucose estimation, improving clinical trust and model transparency.

**Strengths:**

- Accurately predicts glucose levels from noninvasive ECG data in real-world, ambulatory settings.
- Effectively handles class imbalance and improves hypoglycemia detection via a novel composite loss function.
- Robust to missing data (>30% beats) due to Transformer-based temporal modeling.
- Outperforms multiple baseline models in both regression accuracy and clinical safety (Clarke Error Grid).

**Summary Of The Paper:**

This paper introduces ECGluFormer, a deep learning model that noninvasively estimates glucose levels from single-lead ECG in free-living conditions. It addresses two key challenges: under-reporting of hypoglycemia due to class imbalance and signal loss from motion artifacts. The model uses a beat-level encoder with a multi-objective loss function that emphasizes low glucose levels and a Transformer to aggregate predictions while preserving temporal patterns, even with >30% missing beats. It significantly outperforms five baselines across regression and classification tasks, particularly in hypoglycemia detection, by effectively capturing glucose-relevant features like HRV from ECG morphology. Overall, looks like a well executed work!

**Weaknesses:**

- Requires synchronized ECG-CGM data for training, which may limit generalizability to new datasets.
- Lacks interpretability study regarding which ECG features correlate with glucose levels.

---

### Official Review · Reviewer_JE7X · 2025-07-18
**ECGluFormer: A Transformer-Based Deep Learning Model for Non-Invasive Glucose Prediction from Intermittent ECG in Free-Living Conditions**

**Confidence:** 4
**Clarity Of Writing:** good
**Clinical Significance:** great
**Methodological Novelty:** great
**Overall Rating:** 6

**Experiments And Results:**

good

**Questions For The Authors:**

1. Subject-specific vs. general models: Could you clarify whether ECGluFormer can be adapted for population-level modeling or rapid personalization with few samples? A discussion or experiment on cross-subject generalization would influence my score positively.
2. Positional encoding justification: How sensitive is the model to the choice of sinusoidal encoding vs. learned embeddings? Have you compared the Transformer aggregation against temporal convolutional models (e.g., TCN)?
3. Clinical interpretability: Can you visualize which ECG beats or morphological features (e.g., QT interval) contribute most to hypoglycemia predictions? Adding this would strengthen the case for real-world deployment.
4. Impact of RMSE: While your proposed loss boosts sensitivity and Zone D accuracy, RMSE is notably worse than with MSE. Have you tested hybrid losses that preserve regression accuracy while improving detection metrics?

**Strengths:**

1. Clear clinical relevance: The model addresses CGM limitations in hypoglycemia detection, a critical safety issue in diabetes care.
2. Novel loss design: The proposed multi-objective loss function effectively tackles class imbalance and improves predictions in the hypoglycemic range (Zone D).
3. Transformer-based aggregation: Modeling beat timing explicitly with time-based positional encodings offers a principled way to handle missing ECG beats and captures dynamic patterns like HRV.
4. Comprehensive evaluation: The model is rigorously evaluated on both regression and classification metrics, and compared against strong baselines.
5. Strong empirical results: ECGluFormer achieves the highest Zone A+B coverage (98.99%) and lowest Zone D error (1.00%) across subjects, outperforming all baselines in RMSE and sensitivity.

**Summary Of The Paper:**

The paper introduces ECGluFormer, a novel deep learning framework for non-invasive glucose prediction using single-lead ECG signals. The model addresses two key challenges in ECG-based glucose monitoring: (1) hypoglycemia under-reporting due to data imbalance, and (2) signal intermittency in free-living settings. ECGluFormer consists of a beat-level encoder (based on InceptionTime CNN) and a Transformer that aggregates beat embeddings over 5-minute intervals using time-based positional encodings. The authors also propose a multi-objective loss combining normalized MSE, a distribution loss, and inverse-frequency weighting to emphasize performance in low-glucose ranges. Extensive experiments on the PhysioCGM dataset demonstrate superior performance over five baselines across metrics including Clarke Error Grid zones, RMSE, sensitivity, and specificity.

**Weaknesses:**

1. Per-subject modeling limits generalizability: Each model is trained separately per subject, which raises concerns about scalability and generalization to unseen individuals. A cross-subject or personalized transfer learning setting would strengthen the clinical utility.
2. Limited interpretability: While the architecture captures beat timing and morphology, insights into physiological features or attention mechanisms that drive glucose predictions are not provided. Including such interpretability (e.g., attention heatmaps or salient waveform patterns) would improve trustworthiness.
3. Lack of real-world deployment discussion: The paper would benefit from a discussion on runtime feasibility, model latency, or integration with wearable ECG devices (e.g., smart patches or Holter monitors).
4. RMSE trade-off: While the model improves clinical metrics (Zone D, sensitivity), it suffers higher RMSE compared to simpler losses like MSE. Clarifying the clinical impact of this trade-off would help contextualize this sacrifice in point-wise accuracy.

---

### Official Review · Reviewer_ujs6 · 2025-07-18
**Novel estimation of glucose levels from ECG**

**Confidence:** 5
**Clarity Of Writing:** excellent
**Clinical Significance:** excellent
**Methodological Novelty:** great
**Overall Rating:** 8

**Experiments And Results:**

excellent

**Questions For The Authors:**

What is the memory/compute footprint of inference, and do you foresee real‑time execution on current smartwatch chipsets without cloud off‑load?

Could periodic finger‑stick calibrations be fused with ECGluFormer outputs to form a closed‑loop “CGM‑light” system that reduces cost while maintaining safety?

Attention weights highlight which beats matter most. Have you inspected whether the model attends to QT‑interval prolongation or HRV changes that clinicians recognize?

**Strengths:**

This study takes an important step toward needle-free, continuous glucose insight by converting a single-lead ambulatory ECG stream into clinically interpretable glucose estimates, emphasizing hypoglycaemic ranges where current CGMs can falter and patient safety risks are highest. The architecture’s two-stage design (beat-level CNN + time-aware Transformer) and composite loss that up-weights rare low-glucose samples are smart choices that improve coverage of clinically critical events while keeping the model compact enough to envision eventual on-device use. Robustness testing—randomly masking large portions of beats to mimic motion artefact—demonstrates resilience that matters for real-world wearables outside controlled lab settings. Performance reporting in terms familiar to clinicians (Clarke Error Grid zones, RMSE) supports translation by letting providers quickly benchmark accuracy against established CGM expectations. Finally, demonstrating strong accuracy in subject-specific models suggests a viable path to rapid personalization—aligned with precision-health goals in diabetes self-management and potential integration into decision-support workflows.

**Summary Of The Paper:**

ECGluFormer is a two‑stage deep‑learning pipeline that turns a single‑lead, ambulatory ECG into a minute‑by‑minute glucose estimate. A beat‑level CNN encodes 3‑s waveforms; a lightweight Transformer with time‑aware positional encoding aggregates those beats over each 5‑min continuous‑glucose‑monitor (CGM) interval. A custom composite loss (distribution alignment + normalized MSE with inverse‑frequency weighting) specifically up‑weights rare—but clinically critical—hypoglycaemic samples.
Validated on the PhysioCGM data set (10 adults with type‑1 diabetes, up to 17 days of free‑living data), subject‑specific models achieved 98.99 % of predictions in Clarke Error Grid zones A+B and only 1 % in zone D, outperforming five classical and neural baselines while remaining robust when ~34 % of beats were dropped to mimic motion artefacts. The results are discussed in the context of type-1 diabetes.

**Weaknesses:**

Generalizability remains the biggest hurdle: the dataset is small (10 adults with type 1 diabetes) and relatively homogeneous, so broader demographic, physiologic (e.g., arrhythmias), and behavioral diversity will be needed before clinical deployment. Because each model is trained per participant, scalability to new users is uncertain; exploring population pretraining with lightweight calibration (transfer learning, meta-learning, or federated personalization) could greatly improve deployment feasibility. The composite loss that enhances low-glucose detection sometimes increases overall RMSE—an issue regulators and clinicians will scrutinize—so an ablation or adaptive weighting strategy that balances safety-critical lows with overall accuracy would be helpful. Reliance on chest-based ECG alone may limit adherence over long wear periods; fusing additional low-burden signals (wrist PPG, accelerometry, intermittent finger-stick calibration) could bolster robustness when ECG quality drops. Lastly, the manuscript would benefit from an outline of the translational pathway: prospective real-time validation, human-factors testing for alerts, integration into insulin-dosing decision support, and consideration of regulatory requirements for Software-as-a-Medical-Device clearance across adult, pediatric, and special populations (e.g., pregnancy).